# Parallelization of Array Method with Hybrid Programming: OpenMP and MPI

Apolinar Velarde Martínez 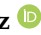

Departamento de Sistemas y Computación, Instituto Tecnológico El Llano Aguascalientes,
Carr. San Luis Potosí Aguascalientes Km. 18, El Llano 20330, Aguascalientes, Mexico;
apolinar.vm@llano.tecnm.mx

**Abstract:** For parallelization of applications with high processing times and large amounts of storage in High Performance Computing (HPC) systems, shared memory programming and distributed memory programming have been used; a parallel application is represented by Parallel Task Graphs (PTGs) using Directed Acyclic Graphs (DAGs). For the execution of PTGs in HPC systems, a scheduler is executed in two phases: scheduling and allocation; the execution of the scheduler is considered an NP-complete combinatorial problem and requires large amounts of storage and long processing times. Array Method (AM) is a scheduler to execute the task schedule in a set of clusters; this method was programmed sequentially, analyzed and tested using real and synthetic application workloads in previous work. Analyzing the proposed designs of this method in this research work, the parallelization of the method is extended using hybrid OpenMP and MPI programming in a server farm and using a set of geographically distributed clusters; at the same time, a novel method for searching free resources in clusters using Lévy random walks is proposed. Synthetic and real workloads have been experimented with to evaluate the performance of the new parallel schedule and compare it to the sequential schedule. The metrics of makespan, waiting time, quality of assignments and search for free resources were evaluated; the results obtained and described in the experiments section show a better performance with the new version of the parallel algorithm compared to the sequential version. By using the parallel approach with hybrid programming applied to the extraction of characteristics of the PTGs, applied to the search for geographically distributed resources with Lévy random walks and applied to the metaheuristic used, the results of the metrics are improved. The makespan is decreased even when the loads increase, the times of the tasks in the waiting queue are decreased, the quality of assignments in the clusters is improved by causing the tasks with their subtasks to be assigned in the same clusters or in cluster neighbors and, finally, the searches for free resources are executed in different geographically distributed clusters, not sequentially.

**Keywords:** shared memory programming; distributed memory programming; hybrid programming; high performance computing systems; clusters; array method; parallel task graphs

## 1. Introduction

The needs for increased speed for data processing and storage of large volumes of information generate the construction and use of High Performance Computing (HPC) systems of parallel architectures. The use of different parallel architectures will continue into the foreseeable future, broadening the available range of hardware designs even when looking at high end systems [1]. We resort to multi-core architectures (parallel computing) for minimal overall completion time [2], to improve scientific application performance [3,4] and I/O overheads and communication times [3].

According to [5], multi-core parallel systems are computer clusters, grid computer systems and cloud computing systems. Due to the heterogeneity of their elements and scaling, commodity clusters have become the de facto standard in parallel processing due to their high performance to price ratio; clusters are also gaining in popularity, mainly

under the assumption of fast interconnection networks and memory buses and can be thought of as a hierarchical two-level parallel architecture, since they combine features of shared and distributed memory machines [2].

For the operation and exploitation of clusters in cooperative work, load balancing, searches for free resources, resource planning and optimization of assignments of PTGs to Processing Elements (PEs), it is necessary to program sequential software systems and parallel systems.

For resource scheduling and optimization of PTG allocations to resources in an HPC system, a scheduler is required. A scheduler divides its execution into two phases, scheduling and allocation of resources; in [6], the scheduling phase is in charge of searching for the best assignments of resources to tasks to reduce the waiting time (the time that the tasks wait in a queue), and makespan is defined as the time the task spends from arrival in the queue until the completion of its execution in the system. The phase of resource allocation is the allocation of resources to the PTGs after scheduling.

The scheduling is performed in two phases; in first phase, a search for free resources in all clusters is carried out to determine the positions of the PEs that remain free; second a search for the best allocations to optimize the execution times of the tasks is performed. When the search for free resources occurs, large amounts of processor time and network resources are required and for the search for the best allocations of resources to PTGs a long processing time and a large amount of server memory storage is required.

The scheduling problem has motivated the development of a plethora of research works, whose solution proposals are based on the programming of heuristic and metaheuristic algorithms, as well as emulations in computational environments of scale-invariant phenomena observed in biological systems, with different high level programming languages. Although sequential programming has been the de facto programming, variants of parallel programming have been proposed over the years to address this problem.

When developing parallel programming to parallelize an application, either autoparallelization or user-developed parallel programming [4] can be used. In user-developed parallel programming, the program is composed of data declarations, assignments and control flow, following the syntax and abstractions of a programming language; the parallelism is related to physically performing tasks in parallel [7].

There is an active research interest in hybrid parallel programming models, e.g., models that perform communication both through message passing and memory access [2]. The mixing of shared memory and message passing programming models within a single application has often been suggested as a method for improving scientific application performance on clusters of shared memory or multi-core systems [3]. The hybrid MPI–OpenMP programming model intuitively matches the characteristics of a cluster of SMP nodes, since it allows for a two-level communication pattern that distinguishes between intra- and inter-node communication [2,8]. Similarly, this type of model exposes most coordination issues such as problem decomposition, communication and synchronization to the programmer, although low-level models offer extensive tuning opportunities for expert programmers at the cost of significant effort [9,10]; this programming, also considered as low-level abstraction and explicit control, enables manual optimization and may result in high performance on a single architecture [10]. These languages are heavily used in the industry [10,11].

The development of this research work focuses only on converting a sequential application to a parallel application [12], taking as an example some references in the literature that have made this conversion [3,13–15]. The work focuses specifically on the conversion of the array method for task scheduling using the array method [6] (for a broad reference to the literature on task scheduling, the reader can consult [6]), with the search for free resources through Lévy random walks considering the works of this research area [16–20] and the design of the parallel application with existing criteria in the literature [4,21–23]. Converting a sequential application to a parallel application is a common practice to speed up

application performance and exploit multi-core architectures as well as make applications more energy efficient.

Task planning is an application that, due to its intrinsic nature of being an NP-complete (non-polynomial) problem [24], can be approached in its two phases (planning and assignment) through architectures that support the development of parallel software systems with languages such as C and C++ and the OpenMP and MPI libraries, because they produce large execution times.

Research works related to task planning in HPC systems do not carry out the search phase for free resources and use different mechanisms to identify them. In this work, we do not overlook this process, and Lévy's random walks are proposed as a new mechanism that performs complex searches in computational resource environments while attending to the different investigations in the design of parallel applications.

As mentioned, in this research work the parallelization of the array method proposed in [12] is presented using hybrid programming OpenMP and MPI in a cluster of servers. The parallelization of the system has been divided into the parallelization of the scheduling phase and the parallelization of the allocation phase. The scheduling phase searches for the best assignment of the resources of the clusters to the tasks, with the use of the UMDA algorithm, and the parallelization of the allocation tasks to the resources phase, which uses a search for resources in the different clusters of the HPC system using Lévy random walks, also mentioned in this work as Lévy flights.

### 1.1. Justifications of This Research Work

The design and construction of this research work is justified by the contributions made to improving the performance of the scientific application, improving I/O overheads, improving communication times and the search for free resources in an HPC system. The following points explain the justification for this work.

- The sequential method proposed in [6] is designed to be scaled to a parallel method that improves performance and optimizes the use of the resources of the cluster where it is executed. Therefore, in this work a parallel system for the phases of planning and assignment of resources to tasks is described and the experiments carried out are presented and described.
- The I/O overheads are improved with the parallel system as experimentations show; times of the metrics evaluated are better with the parallel system compared to the sequential system.
- In this research work, the architectural network design (configuration) of a cluster-class HPC system for the free movement of a mobile software agent for the search of free resources, using Lévy flights, is proposed; this will result in speeding up the completion of tasks that require execution times on the HPC system.
- The design of the parallel application for task scheduling and the allocation of computational resources to tasks is presented; this work serves as a basis for future work in the area of the development of parallel algorithms for task scheduling in HPC systems.

### 1.2. Pros and Cons of the Research Project

In this section, we state the pros and cons of developing this research project. We emphasize that it is a research project that is currently being developed within the university, so the results presented in this work are the experiments carried out within the university campus; active research is continuing to test the proposed system in more realistic environments.

Pros:

- The functionality in a cluster of servers with the Linux operating system.
- Improving the performance of the array method in a high performance computing system with the use of centralized memory programming and distributed memory.
- Different geographically distributed clusters have been used to test an algorithm with a search for resources randomly, not sequentially.

- This project has been proposed to serve as an example for the development of future projects that parallelize applications using hybrid programming, because in this work the parallelization process of a sequential application to a parallel application is described.

  Cons:

- It is an empirical model that has been tested on a limited server cluster, it is necessary to perform tests on cluster systems with more servers.
- The structure of the computer network for the clusters must be ad hoc to the functionality of the project to allow the mobile agent to move in the computer network, which implies in future work standardizing the movement of the agent to any network structure.

This work is organized as follows. In Section 2, basic definitions and terms used throughout this work are defined; Section 3 includes a statement of the problem, which is the specification of the formalization of the task planning problem in an HPC system; in Section 4, some works related to the technologies used in this work are summarized; Section 5 presents a description of the array method scheduler and its data structures; Section 6 presents the way in which Lévy's random walks are used as a search strategy for free resources in this paper; Section 7 describes the flow chart of the parallel design of the array method; Section 8 defines the materials and methods used to carry out experimentations; results are described in Section 9; the discussion on this research work appears in Section 10 and, finally, conclusions are presented in Section 11.

## 2. Basic Definitions

Since this work is based on [6,12], some definitions from those papers appear in this section; furthermore, other definitions are added for this research work. The proposed definitions are referenced in the explanation of the research, the proposed algorithms and the experiments performed.

**Definition 1.** *The target system consists of $C_l$ clusters, $C_1, C_2, \ldots, C_l$ where l is the number of clusters contained in the HPC system. Each cluster contains m heterogeneous processors with n processing cores or resources. Therefore, $C_{l,m,n}$ is cluster k, processor m, processing core n.*

**Definition 2.** *A PTG can be modeled by a Directed Acyclic Graph (DAG) $T = (N, E)$ where:*
*$N = \{n_i : i = 1, 2, \ldots, N\}$ is a set of N vertices or subtasks that constitute the task, and*
*$E = E_{i,j} : i, j = 1, 2, \ldots, N$ is a set of E edges.*
*A PTG can be characterized by*

$$(n_i, \{1 \leq j \geq n_i | T_i\}, G_i, V_i, W_i) \tag{1}$$

*where:*
*$n_i$: the number of subtasks in $T_i$.*
*$\{1 \leq j \leq n_i | T_i\}$ : the set of subtasks.*
*$G_i$: the set of directed relationships between the subtasks.*
*$V_i$: the number of levels of $T_i$.*
*$W_i$: the width of each level $T_i$ represented by an array.*
*The PTG consists of a set of N nodes and a set E of edges (directed relationships). The nodes represent execution requirements of the task T. The requirement of each task is represented by $w_{n_i}$. The directed relationships show the flow of execution of subtask $\tau_{i,j}$ to subtask $\tau_{i,k}$ of task $T_i$. Each of the terms used in this definition is explained below.*

**Definition 3.** *PTG density is the number of links between PTG subtasks. It is represented by:*

$$|G_i|. \tag{2}$$

*The communication percentage of a task is obtained by counting the total number of edges and the number of subtasks of the PTG.*

**Definition 4.** *A synthetic workload is constituted by a set of DAG tasks and is denoted by:*

$$S = T_1, T_2, \ldots, T_n \qquad (3)$$

*Each $T_n$ consists of a random number of subtasks and a random number of edges. n is defined in the workload generation algorithm.*

**Definition 5.** *Let us consider Definition 2 from [12]; it follows that each PTG will request $\eta_i$ resources, which must be extracted from the scheduler's resource array, then the next condition is established:*

$$\forall \, \eta_i \; \exists \; R_{l,m,n}, \qquad (4)$$

*for the PTG to complete its execution.*

**Definition 6.** *A Lévy flight is a type of random walk in which the increments are distributed according to a "large tail" probability distribution. Specifically, the distribution used can be approximated by a power law of the form [20]:*

$$P(l) \; \propto \; l^{-\mu} \; with \; 1 < \mu \leq 3 \qquad (5)$$

*where µ is a constant parameter of the distribution known as the exponent or scaling parameter.*

**Definition 7.** *Consider a software agent $SA$, which locates any idle $R_{l,m,n}$ found in any $C_l$ using Equation (2), then registers it in an array of resources, and Condition (1) is satisfied.*

### 3. Problem Statement

Consider a scheduler $S$ in a set of clusters $C_l$, let a function $f$ schedule and assign workloads $W$ constituted by tasks, and a software agent search a set of free resources on an HPC system represented by SA; then $f$ is to minimize the number of unallocated resources in the target system, minimize the number of unassigned tasks and minimize the number of unused resources of the clusters, so the objective function is given by:

$$min f = \sum_{i=1}^{n} C_{l,m,n} + \sum_{i=1}^{n} C_{l,m,n} + \sum_{i=1}^{n} C_{l,m,n}. \qquad (6)$$

### 4. Related Works

In the following paragraphs, the works related to the topics used in this research work are addressed. The related topics have been divided into works that present a conversion from a sequential application to a parallel one, works that have been developed with OpenMPI and MPI, works in the area of scheduling tasks with search and use of resources, research work on applications and theories developed in Lévy flights and, finally, works addressing parallel application designs.

Several works of the literature present a conversion from a sequential application to a parallel application [13–15]. These conversions improve the acceleration and efficiency of the application [13], looking for the acceleration of the results [14], to improve performance of an acceleration on multi-core architectures and to make the application more energy efficient [15].

When we decide to change the sequential programming to a parallel programming, and also the development is decided to be done in a multi-core system, OpenMP is considered the de facto standard when shared memory parallel programming is used, because it provides a simple but very powerful method of specifying shared work between threads [3]. OpenMP extends the base programming languages: Fortran, C and C++, and is an open specification for shared memory parallelism [4,25]; its success is based on providing a simple and flexible model for developing parallel applications on different platforms ranging from embedded systems and device accelerators to multi-core computing systems [25].

On the other hand, MPI [26] is the de facto standard for the parallel programming of distributed memory. MPI is a set of libraries used as a power tool to communicate a set of servers. MPI libraries tend to be highly optimized for message passing communication [2]. It is not a trivial task to determine the optimal model to use for some specific application [1]. According to [2], the hybrid models [8] can in some cases be more beneficial compared to the monolithic pure message passing model, as they better exploit the configuration characteristics of a hierarchical parallel platform, such as an SMP cluster.

In research works on scheduling and allocation of tasks in an HPCS, it is assumed that there is a set of resources available in a resource pool as in [27,28]; in [28], discovery mechanism searches of resources are used and return the addresses of the resources that match with the provided descriptions of required characteristics; in [29], resources are scheduled prior to path setup requests, and the allocator has resources' metadata [30]; in [31], an assumption is made: each user contributes a certain number of machines (resources) to the common pool of machines (resources) in the cloud; in [32], only a resource management system is proposed, this system manages a pool of processing elements (computers or processors) which is dynamically configured (i.e., processing elements may join or leave the pool at any time); [33] establishes a model where capabilities of the machines are known such as disk space, number of cores, amount of memory and host OS that is running, and in the same way supposes that a large data center and cloud systems already have significant monitoring tools that provide near-real-time updates of various systems to their controllers. In [34], the resources are available on demand, charged on a pay-as-you-go basis, and cloud providers hold enormous computing resources in their data centers while, on the other hand, cloud users lease the resources from cloud providers to run their applications; in [35], the cloud service end user is proposed to use the entire stack of computing services, which ranges from hardware to applications. In [36], in a cloud of resources where tasks are scheduled, the resources are heterogeneous and characterized by power and cost constraints.

Resources in the HPC system scheduler are used to improve metrics such as throughput [28], waiting time [5,27,37], overall average task response time [32] and load balancing [38], therefore the resource search phase, the resource management phase and the use of resources must be analyzed, designed, elaborated and tested perfectly in the computer system.

Considering the importance of searching for resources in an HPC system, in this work we propose Lévy random walks as a search strategy for free resources in the clusters of the target system. Lévy random walks [16–18] is an area of research in various disciplines from ecology to physics [20], and a special class of random walks whose stride lengths are not constant but are selected from a probability distribution with a power law [16,17,19,20]; in [19], the hypothesis that Lévy random walks are optimal when exploring unpredictably distributed resources is established; similarly, in [18] it was pointed out that in Lévy flights divergence of the variance produces long jumps and typical trajectories are self-similar in all scales, showing jumps that are closely interspersed. In the sections below, we explain how Lévy random walks are used to search for free resources in the clusters of the target system.

For the design of a parallel application, different research works [4,21–23] propose a set of steps or criteria, which are: the specification of the hardware or architecture where the application is executed, the parallelization model to be used, the programming methodologies, the workload partition scheme, the implementation strategy, the synchronization, the software libraries and the communication model. Each criterion is a specification that the programmer can develop during the design of the parallel system. According to the aforementioned details, for the development of the parallelization of the array method, the following criteria were used in this research work.

- The specification of architecture. A single microprocessor computing system with multiple cores and shared cache memory has been chosen for the design of the parallel program; according to [21], multi-core systems are predominant in the commercial market and the processing they perform can be implemented in HPC system archi-

tectures, in personal computers and low-cost workstations using standard software components.

- Programming methodology. In this phase, the shared memory programming method has been chosen. According to the structural design of the sequential program, the use of threads for the cycles of the array method has been the most appropriate way to improve the performance of the parallel program. In Section 5, the array method is described and the data structures used are specified, which are arrays controlled by cycles.
- The workload partition scheme. The scheduling methodology allows dividing the program's workloads; in this criterion, the regions of the sequential code with the highest consumption of processing time and that are feasible to parallelize were identified with the GNU gprof tool [39]. These code segments were identified as the loops that update the arrays of the method.
- Software library. Once the cycles of the sequential code with the greatest time consumption were identified, it became necessary to exploit parallelism at the loop level [21,39]; this type of parallelism is an important part of OpenMP applications that contain a lot of data and computer intensive parallel loops. OpenMP applications typically run on HPC platforms that are increasingly complex, large and heterogeneous, and exhibit massive and diverse parallelism [40]. Due to this, OpenMP was chosen as the software library, to allow the multi-threaded execution of the parallel program.

## 5. Array Method

The array method [12] is a scheduler developed in C language, sequentially. The main function of this scheduler is to schedule tasks on an HPC system. The functionality of the method is based on a set of four arrays called resources matrix, characteristics matrix, assignment matrix and start times matrix of the tasks. For each array, iterative processes or loops are executed. Iterative processes are parallelized with a master, which spawns additional threads to cover all iterations of these iterative processes (one thread per core). The parallelization process is described in the Section 5.1.

### 5.1. Parallelization Process

The parallelization process makes use of OpenMP data parallelism and pragmas that are based on shared memory. The following steps were followed:

- Definition of shared and private variables. The scopes of the variables shared by all the functions of the method were defined, as well as all the private variables that each function uses.
- Concurrences in the program. The concurrences are carried out by declaring the critical sections to ensure mutual exclusion in the execution of the code blocks and thus prevent the simultaneous execution of the code segments by different threads.
- Finally, using explicit parallelism, for the distribution of parallel work for each core, each function was distributed based on the number of cores each processor contains.

The paragraphs below explain, in a very reduced form, every array that constitutes the array method and the execution form of this scheduler; an extended explanation of the array method and the first part of the parallelization of the method are explained in [6,12]. The objective of explaining each of the four structures of the array method is to highlight that the creation, use and update of the arrays depend on consecutive cycles that are adaptable to coding with OpenMP libraries.

### 5.2. The Resource Matrix

The resource matrix stores the characteristics of each Processing Element (PE) of the clusters; a verification cycle of the cluster resources is carried out by a daemon in the background. Updates to this matrix occur when a new element is added to one of the clusters or a cluster is added to the HPC system.

The characteristics of each PE that this array contains are: the number of cores and the distances between the PE and each of the Processing Elements (PEs) of the HPC system. Once a new resource is located, a loop to extract the characteristics of the resource and calculate the distances from the processing element to each of the PEs is executed. In this way, the resource is added to the resource matrix for use.

This array is responsible for storing the state of the resources that are busy or idle. If the resource status is busy, a task is using the resource, and an inactive status of a resource indicates that the resource has not been assigned to a task or has been freed by the completion of the execution of a task.

### 5.3. Characteristics Matrix

PTG characteristics such as PTG identification number, number of PTG routes, number of PTG levels, number of vertices per level and PTG density are stored in the characteristics matrix. This feature extraction process is performed using the depth search algorithm (DFS algorithm) [41].

### 5.4. Assignment Matrix

The assignment matrix is responsible for storing the values produced by each assignment (task to resource). To assign PTGs in the characteristics matrix, two search criteria are used based on the density of the DAG: if the DAG density is greater than the established threshold, Criterion 2 is applied, otherwise Criterion 1 is used.

**Criterion 1.** Once paths in the PTG are determined, the critical path is defined; then, this path will have the highest allocation priority. After this, a pruning process for the PTG is performed once each route is assigned. Cores of each selected PE are considered for every path assignment; the divide and conquer method is applied recursively if the PTG path is greater than the number of cores of the selected PE.

**Criterion 2.** Considering the levels of the PTG and the number of subtasks per level of the PTG, the resource allocation process is carried out. The number of subtasks per level is the factor for finding the processing cores in the PEs to allow the processing data to flow in the same direction. If a PE with a number of cores equal to or less than the number of tasks per level is not found, the divide and conquer method is applied iteratively.

### 5.5. Generation of Populations to Obtain Solutions

The matrix of assignments allows for building the matrices of the solutions that will iterate in the UMDA algorithm. Each solution matrix contains the Hamming distance, the state of the processors and the distance to the target group.

### 5.6. Umda Algorithm

The UMDA algorithm is executed as follows to generate the best assignment of tasks to resources.

Step 1. The solutions of the assignment matrix represent the initial population and the solutions generated afterwards represent the populations that the algorithm uses for the iterations.

Step 2. Individuals are selected from the population generated by the standard truncation method to select half of the initial population; the selection uses the minimization function of the values of three parameters: Hamming distance, state of the processors and the distance to the target group; a tie between the values when evaluating the individuals is resolved probabilistically.

### 5.7. The Matrix of Task Start Times

With the best assignments for the PTG calculated with the UMDA algorithm, the start times are obtained for each of the subtasks of the PTG. Task execution start times are stored in the task start times matrix.

## 6. Lévy Random Walks as a Free Resources Search Strategy

In this section, Lévy random walks are defined as strategy for free resources search. The exponential increase in step length gives the Lévy random walks the property of being scale invariant, and are used to model data exhibiting clustering. Lévy flights are a special class of random walks whose step lengths are not constant, but are selected from a probability distribution with a power law [16,17,19,20].

The random walk is represented by a succession of random variables $X_n$ with $n \in N$ known as a discrete stochastic process. The sequence $X_n, n \in N$ forms infinite sequences $X_0, X_1, \ldots, X_i$ with $X_i \in Z$. That is, if a run starts at state 0, at the next time the agent can move to position $+1$, with probability $p$, or to position $-1$ with probability $q$, with $p + q = 1$. Position $+1$ is considered the PE to the left of the current PE and position $-1$ is considered the PE to the right of the current PE.

This research work proposes the case of a software agent which starts from a specific cluster and PE, and moves in stages or steps along the target system; at each step, it moves a unit distance to the right or to the left, with respectively equal probabilities. For the movement of the agent, let $\zeta_n$ be the *nth* motion of the agent and $P$ be the probability function, then,

1. $P(\zeta_n = +1) = p$, the agent moves one PE to the right of the current position.
2. $P(\zeta_n = -1) = q$, the agent moves one PE to the left of the current position.

$\zeta_n$ are considered as the independent random variables.

The agent moves $k$ steps in total, before registering idle resources in the resource matrix (see Section 5). For these experiments, Lévy flights are used as a search procedure for idle resources. The steps for the execution of the software agent are explained on next section (a reduced algorithm is provided in Algorithm 1).

*Search for Idle Resources in HPC System Using Lévy Random Walks*

For the resource allocation in HPC systems, most research works (as Section 4 explains) assume a centralized resource manager that has a complete vision of network topology, as well as networking and computing resources status; this assumption is not valid for large-scale worldwide grid networks; practically, grid network comprises geographically distributed heterogeneous resources interconnected by multi-domain networks [29]. In this paper, before performing the scheduling and allocation of resources, a resource search engine is proposed in HPC systems, using Lévy random walks to extract the characteristics of each processing element, assuming that ignoring computational resource capacity and availability may affect the overall performance significantly, especially in computationally intensive applications [29].

To generalize the problem, let us consider Figure 1, which has a set of geographically dispersed clusters linked by wireless and wired communication; Figure 1 is the most similar to the actual architecture used in the experimentations. For reasons of space, details about network bandwidth, multi-domain environments, inter-domain and intra-domain topology are not specified, in addition to how the different domains interact to provide end-to-end connectivity.

In Figure 1, circles represent processing elements, filled circles show occupied PEs, circles filled with lines represent PEs without functionality, while unfilled PEs are idle at time $t + 1$ and can be located by the software agent. Links between processing elements within a cluster and links between clusters are represented by solid lines. The three dots above the solid lines show the possible addition of new clusters to the target system.

The example showed in Figure 1 operates as follows: the resources search in the HPC system using Lévy random walks is shown with a dotted line; the Lévy random walk executed by a software agent using the algorithm shown in Algorithm 1 starts in cluster 1, executes a set of steps for searching the PEs in this cluster and migrates to cluster 2, 3, 4 and 5; in cluster 5, a long jump is executed. The search process is repeated for the time set in the

algorithm; at the end of the random walk, twenty idle resources are identified by the SA, the resource matrix is updated and resources are available for scheduling and allocation.

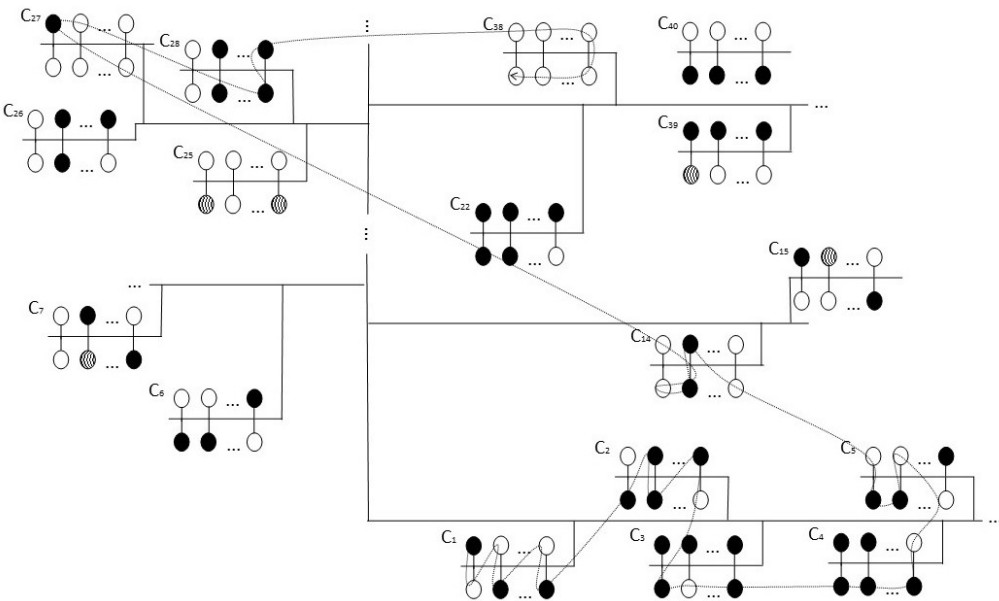

**Figure 1.** HPC systems with clusters showing a search for idle resources using Lévy random walks .

Considering the algorithm proposed in [42], Algorithm 1 shows a reduced structure of the resource search algorithm using Lévy's random walks, operated by the software agent.

---

**Algorithm 1:** Algorithm for free resources search using Lévy random walks.

---

 *Input* : *Starting position for the search in the HPC System*
 *Output* : *List of idle resources in the HPC System*
 *Begin*
  *Assigns the search time by the user;*
  *Identify starting position of software agent in HPC System;*
   *Location of the last search;*
   *Start search from cluster identified as starting cluster*
  *While* (*Search time exists*)
   *Calculate movement of agent using Equation* (2);
   *Identify and verify the status of the EP where the software agent is to be moved;*
   *If* (*PE Status* == *Idle*)
    *Save position and characteristics of the PE;*
   *End_if*
  *End_While*
  *Update resource matrix*
 *end*

---

## 7. Scheduler Parallel Layout

This section describes the flow chart of the parallel design of the array method. Figure 2 depicts the layout of the parallel design of the scheduler algorithm, as an extension to the design proposed in [6]. In the following sections, an explanation of each section of the flow chart is provided.

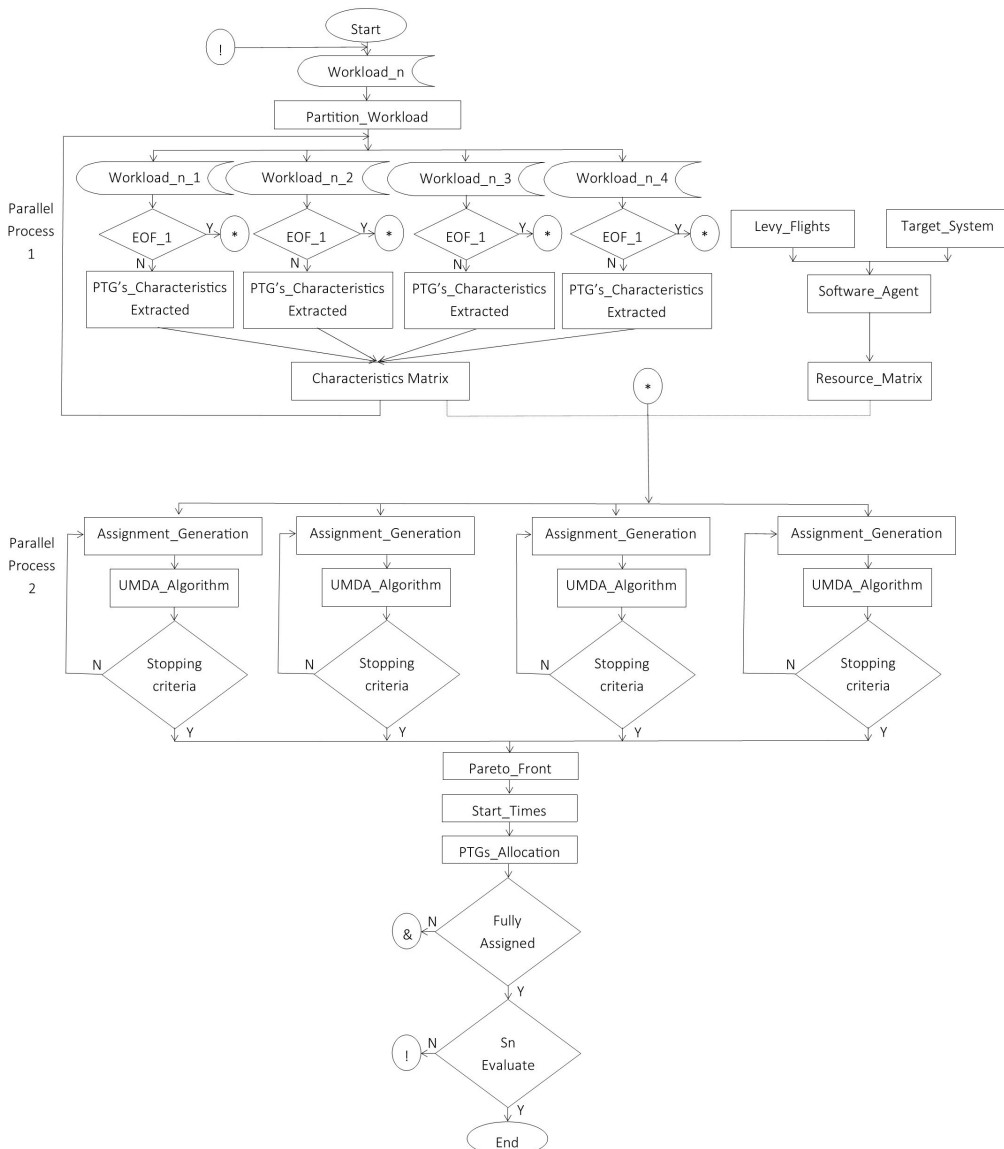

**Figure 2.** Scheduler parallel layout.

### 7.1. Target System

The target system on which the new scheduler algorithm runs is a cluster of 3 servers linked to a high speed network; the cluster stands on Liebres InTELigentes, and Figure 3 depicts architecture of the Liebres InTELigentes server farm cluster. Server "Liebre InTELigente 1" is an Intel Xeon quad core server, responsible for executing the loading and processing of the workloads. The feature and resource arrays are stored on "Liebre InTELigente 2" which is an Intel Xeon dual core server and, finally, the execution of the UMDA algorithm is performed on a third server, "Liebre InTELigente 3" with an Intel Xeon quad core processor.

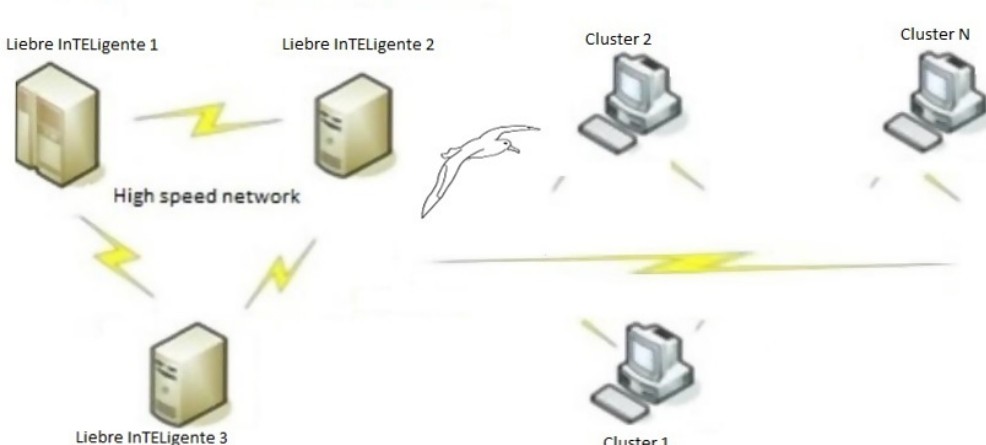

**Figure 3.** Liebres InTELigentes server farm cluster.

The OpenMP libraries allow the execution of the algorithm inside the server; communication between servers is carried out with MPI libraries. The results of the executions are reported and printed on the first server. The search for free resources is carried out using a mobile software agent, which moves in different clusters.

### 7.2. Sections of the Algorithm

The general design of the parallel algorithm is divided into two parallel processes. In the first process, the workload with a set of PTGs is divided by the number of processing cores of the Liebre InTELigente 1 server. Each core is responsible for extracting the characteristics of each PTG (an end-of-file check is executed beforehand). The characteristics of each PTG extracted by each processing core are saved in the characteristics matrix, in the Liebre InTELigente 3 server; the feature array and the resource array are stored on this server.

In the second process, each time a workload completes (all PTGs have been processed), the mapping generation process runs. According to the number of processing cores of the Liebre InTELigente 2 server, the execution of the UMDA algorithm is divided. Every time the execution of the UMDA algorithm finishes in each core, to choose the best assignments of PTGs to the resources, the Pareto front is executed. With the best assignments, the task start time matrix is updated and the tasks are sent to each resource in the HPC system.

A verification of the last processed workload is performed. If the last workload was processed, the parallel system ends, otherwise the algorithm starts its execution again.

The communication between the host servers is done with the MPI libraries. Local processing is done with OpenMP libraries.

For the evaluation of the experiments, both parallel sections (as shown in Figure 2) of the parallel algorithm were verified to detect the parts of the algorithm where the best accelerations are produced in the execution; the first partition called the workload partition allows an acceleration for the extraction of the characteristics of the workload. The second partition, generation of solutions for assigning tasks to resources, allows exhaustive searches with four processing cores. At the end of the search with the processing cores, the Pareto front shows a wide generation of solutions, which allows the choice of better solutions.

The module for searching free resources in the HPC system clusters, called the software agent, runs in parallel to the two processes described above. The software agent simulates the flights of the albatross, and travels in the different clusters of the target system. The albatross appears in Figure 3 on the communication line of the clusters. The Liebre InTELigente 3 server is responsible for executing the flights of the Lévy module.

## 8. Materials and Methods

### 8.1. Hardware

Experiments with the new algorithm and sequential algorithm were performed in the Liebres InTELigentes cluster consisting of 3 servers configured with Fedora and openSUSE Linux operating system: 1 Dell EMC Power Edge Rack Server Intel Xeon generation 2 with 20 cores and 2 Servers HPE ProLiant DL20 Gen10 Intel Xeon. Communication between servers was achieved with Switch Cisco Gigabit Ethernet SG350-28, 28 Ports 10/100/1000 Mbps.

Other considered hardware is 10 clusters constituted by different numbers of process elements. In 5 clusters, the hardware is homogeneous with desktop computers, and 5 clusters present heterogeneous hardware. Experiments were executed with different flavors of Linux operating systems.

### 8.2. Software

The Linux server was used as a resident operating system on servers and clients. The Linux server was responsible for providing the native C language compiler and OpenMP and MPI libraries were manually installed. Additional libraries were installed on the servers to provide communication services in the clusters.

### 8.3. Method Coding

The programming language used in this research work was C using dynamic memory programming and OpenMP and MPI libraries; dynamic memory allows growing and shrinking according to the requirement of synthetic workloads (it was not known how much memory was needed for the program), and memory spaces are used more efficiently.

## 9. Results

In this research work, the experiments were carried out with the makespan, waiting time and quality of assignments performance metrics; the objective of these experiments is to measure the performance of the new parallel algorithm against the sequential algorithm of the scheduler. Additionally, an experiment realized with the software agent, for the search for free resources in the target system, is presented; this experiment is performed with a comparison between a sequential search and a search using Lévy random walks. The results are shown in the following figures and explained in the paragraphs below.

For the makespan evaluation, a division of the workloads was made: loads with a high density of the PTG and loads with a low density of the PTG. In the next subsections, both types of evaluation are explained.

### 9.1. Loads with a High Density of the PTG

The first execution of the experiments consists of workloads that contain a high level of PTG density, that is, the created PTGs contain a large number of subtasks, which makes it necessary to require many resources from the target system. Figure 4 shows the results obtained with this first experimentation. The *X* axis shows the number of PTGs used in each workload, the *Y* axis shows the time that the algorithm consumes (on average) to finish each workload with that number of PTGs. To obtain the time that the algorithm consumes (represented on the *Y* axis), different executions were carried out, so the result that is represented is an average of the executions.

Observations. With a high density in each PTG, the parallel algorithm outperforms the sequential algorithm in all runs. Each part of the parallelized algorithm provides better response time: the partitioning of the workload and the generation of solutions for assigning tasks to resources.

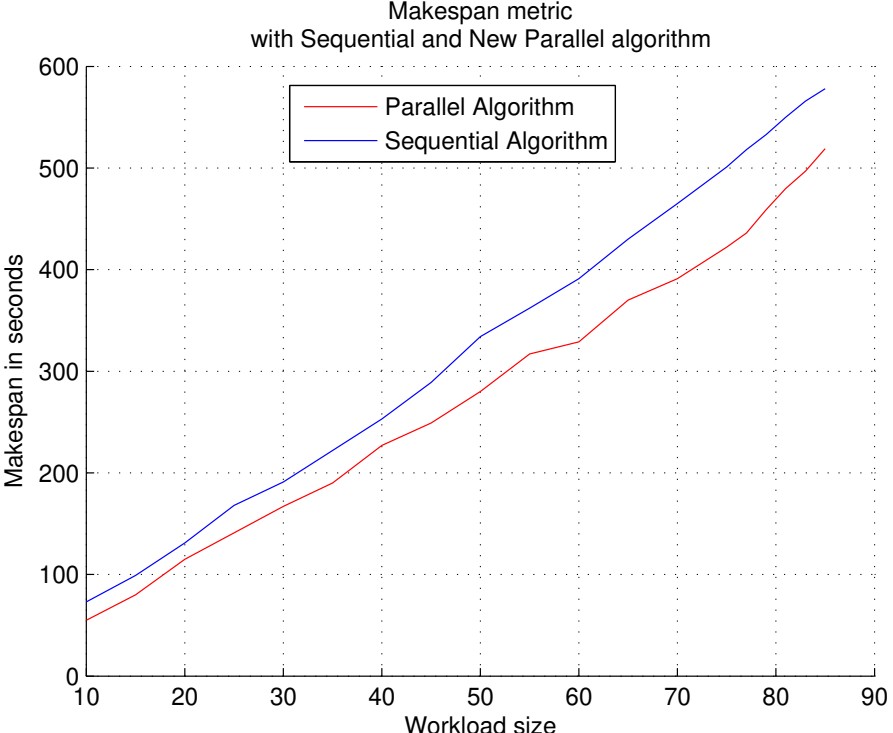

**Figure 4.** Workloads with a high density of the PTG.

### 9.2. Loads with a Low Density of the PTG

A second generation of experiments was carried out with a low density of the PTG, that is, the number of subtasks per PTG is smaller than in the first experimentation, this causes a smaller number of resources to be requested by the PTG in the target system. Figure 5 shows the results obtained with loads with a low density of the PTG.

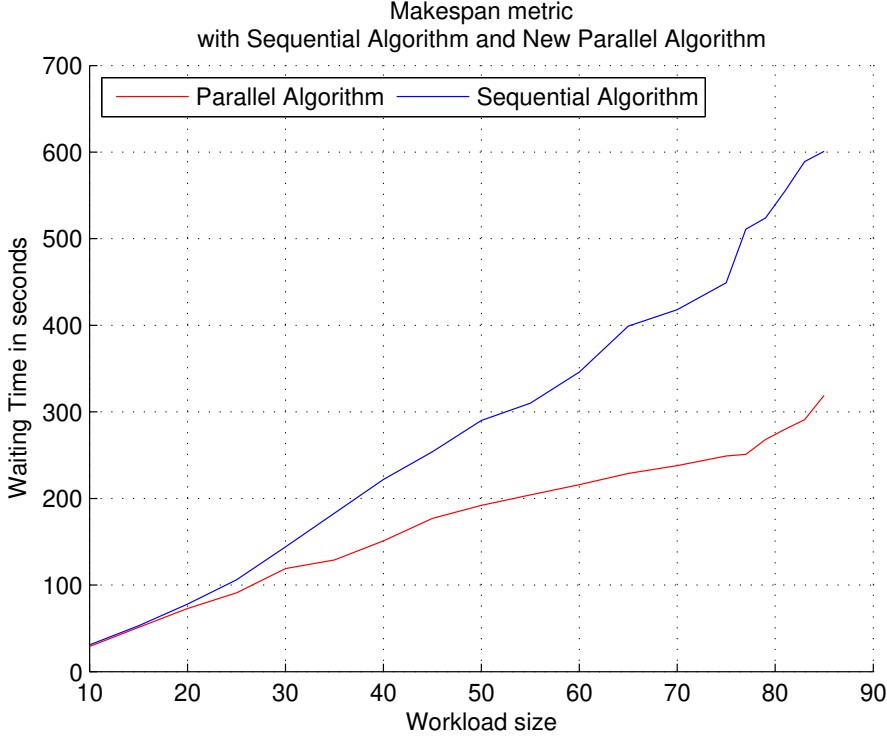

**Figure 5.** Workloads with a low density of the PTG.

Observations. The sequential algorithm is optimal when the number of PTGs in the workload is not large, so both algorithms (sequential and parallel) provide similar performance. A mean difference starts to show when the number of PTGs increases in workloads.

The experiments with the waiting time metric measure the permanence time of the tasks in the waiting queue; two different loads are experimented with using this metric, the first type of workload is with high density tasks (see Equation (6)) and the second type of load is with low density tasks (see Equation (6)). The objective of using workloads with high density is to verify if the algorithm can overcome the starvation of tasks in the queue, since these types of tasks request a lot of system resources; with workloads with low density, it allows verification of whether the clusters can be flooded (filled) very easily by having a powerful free resource search and allocation strategy; high utilization of resources in clusters can cause high levels of communication in the target system.

The following graphs show the results obtained with the waiting time metric. The performance of the parallel algorithm when experimenting with high density workloads obtains a better response time to the tasks during the course of the experiments, i.e., the tasks spend less time in the waiting queue, as shown in Figure 6; however, scheduling tasks using multi-level feedback queue scheduling consumes large amounts of processor time because tasks with many subtasks require a lot of system resources, causing them to be dispatched at different scheduling levels. The performance of the sequential algorithm is partially similar to the results of the parallel algorithm with the first loads, and it does not generate starvation due to its sequential nature, i.e., it attends to each task according to its entry into the queue.

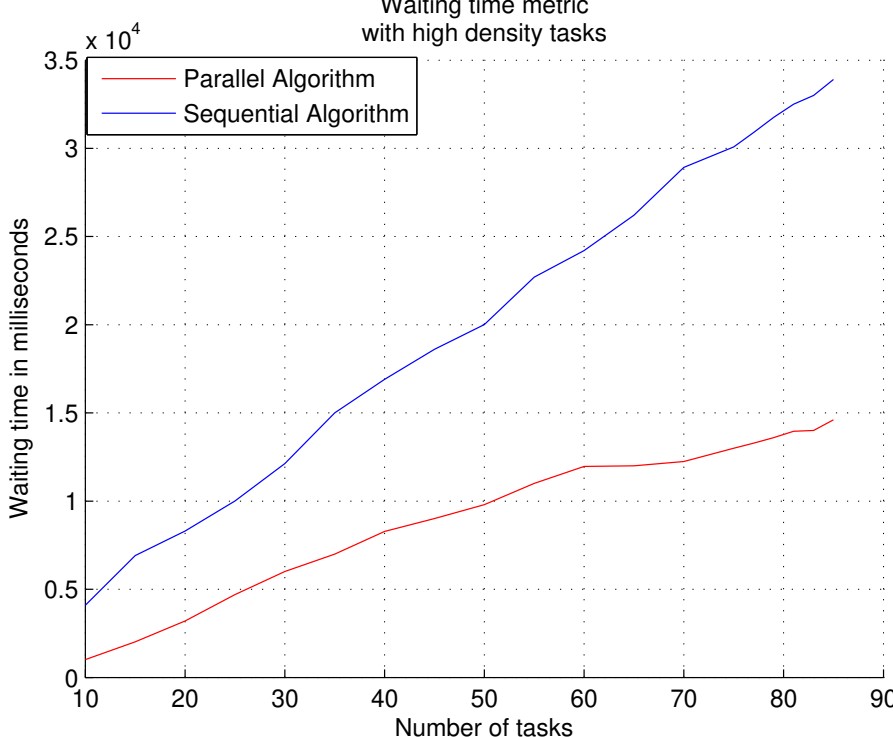

**Figure 6.** Results obtained with the parallel and sequential algorithms using the waiting time metric with high density tasks.

The performance of the parallel algorithm when experimenting with low density workloads obtains the best times in the waiting times of the tasks in the queue, as shown in Figure 7. The low number of subtasks for each task makes the parallel algorithm in coordination with the mobile agent (which runs independently of the scheduling algorithm) easily perform the allocation of tasks in the target system. With the sequential algorithm,

the waiting times of the tasks are increased; the advantage of sequential execution of tasks is a very low probability of starvation.

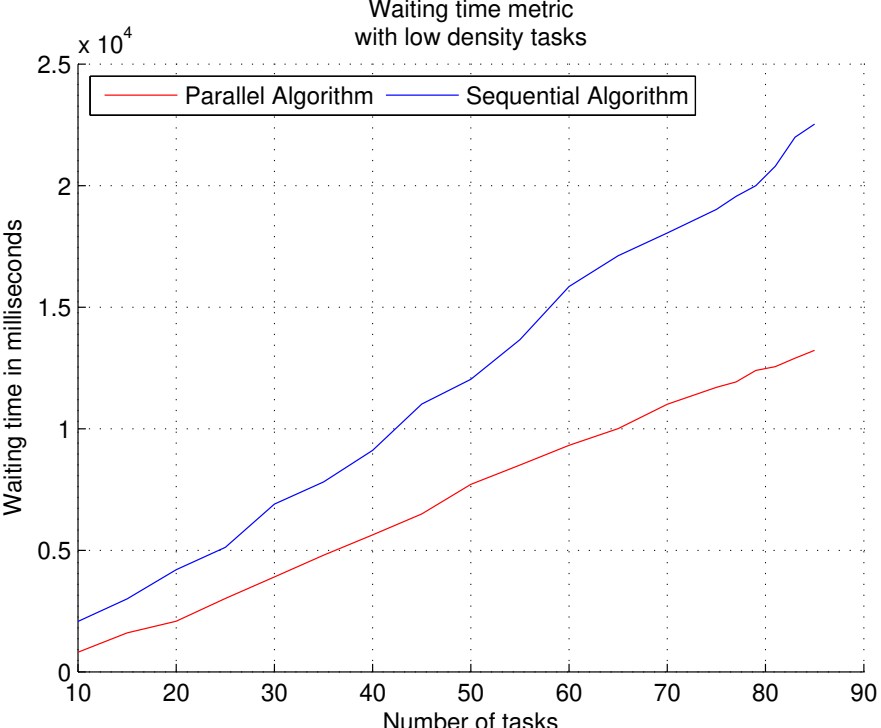

**Figure 7.** Results obtained with the parallel and sequential algorithms using the waiting time metric with low density tasks.

Experiments with the quality of allocations metric are also carried out with two types of workloads: workloads with high density tasks and workloads with low density tasks. Measuring the quality of task assignments aims to determine the spread of tasks in the system. When tasks are spread across multiple clusters, communication costs are very high; data between subtasks are transported long distances within the HPC system to complete the task, and subtasks keep resources busy (but idle) waiting for data. Optimal assignments occur when the same task can be executed within the same cluster or in close neighbor clusters.

The parallel algorithm allows more task assignment searches (in less time) than the sequential algorithm when processing tasks with a high density; deeper searches with the parallel algorithm produce more optimal results with this metric. The parallel execution of the meta-heuristic algorithm, together with the Pareto front technique, allows us to discriminate non-optimal assignments and obtain assignments that produce better results in inter-task communication and the use of free resources. Figure 8 shows the results of the experiments with the performance metric quality of allocations and workloads that contain tasks with high density; the results in the graph show the communication times between the tasks during the total execution of each workload. The parallel algorithm manages to locate tasks with better precision in the target system, optimizing communication during the execution of all the tasks of the workload.

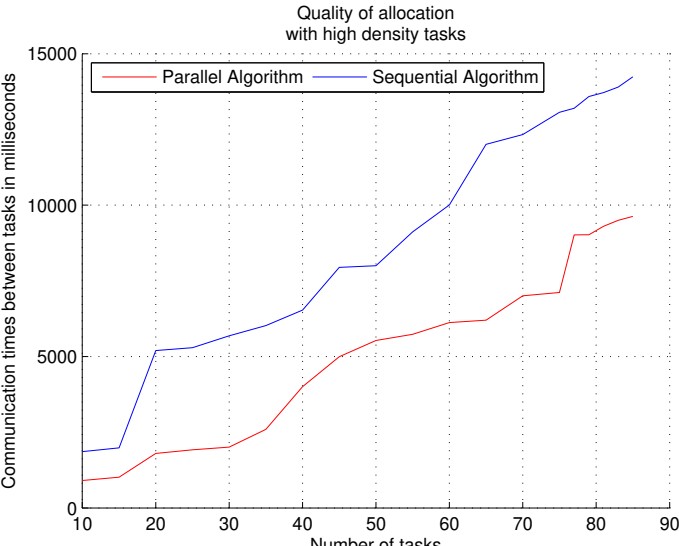

**Figure 8.** Results of the experiments with the parallel and sequential algorithms using the performance metric quality of allocations and workloads that contain tasks with high density.

When using workloads with low density, to measure the performance metric of quality of allocations, the results with the parallel algorithm are superior to the sequential algorithm as Figure 9 shows, when the execution time of the complete workload is measured. The best assignments (allocations) of the tasks in the different clusters decrease the communication times between the tasks and allow assigning (allocation) of more tasks within the target system.

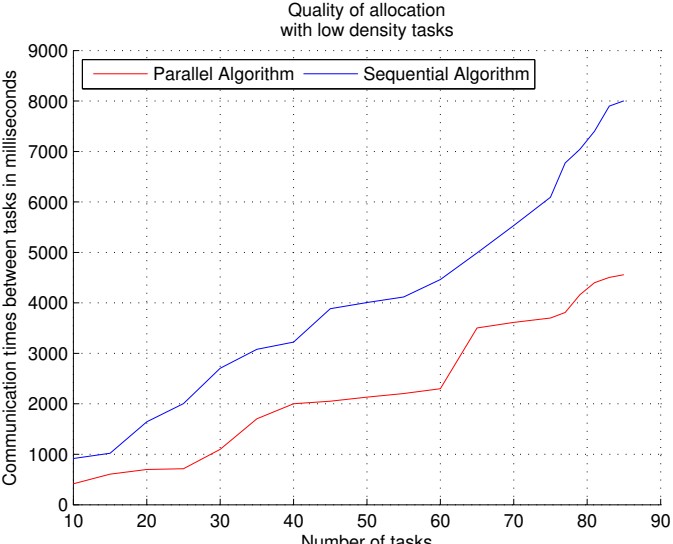

**Figure 9.** Results of the experiments with the parallel and sequential algorithms using the performance metric quality of allocations and workloads that contain tasks with low density.

For the experiments with Lévy random walks, a comparison with the sequential search of free resources was performed. The results are shown in Figure 10. With Lévy random walks, updates to the resource matrix are more frequent, causing more free resources to be available during allocations, which makes the allocation algorithm more efficient and faster.

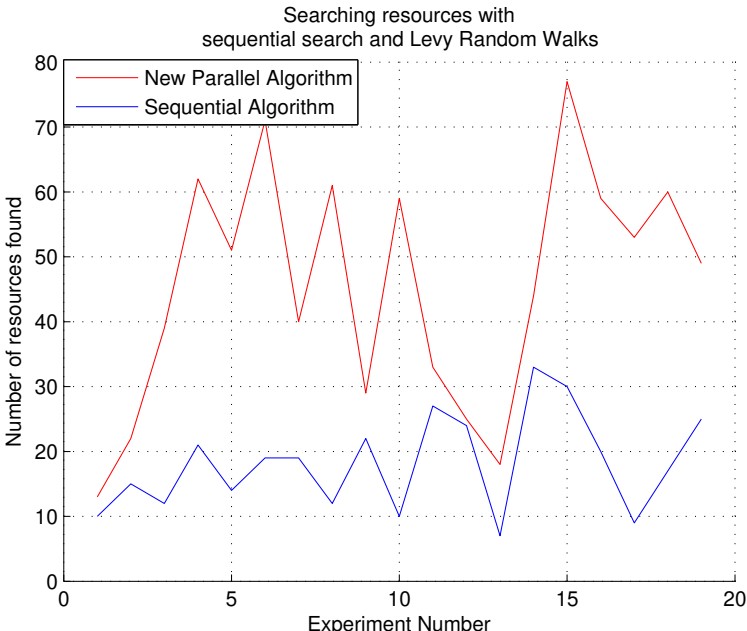

**Figure 10.** Results of the experiments of free resources search with the Lévy random walks and sequential algorithm.

## 10. Discussion

We have proposed a task scheduling system with hybrid programming, which runs on a server cluster from which a mobile agent is sent to different physically disaggregated clusters. The agent makes use of Lévy random walks to optimize its movements within the clusters. The findings found in the comparisons made with the sequential algorithm show the new parallel algorithm has better performance results, better response times with heavy workloads and more efficient searches with the new technique used. As explained in the previous paragraphs implies the use of more efficient technologies, such as multi-core processors, faster communication links and more efficient hardware architectures.

Future research works will be addressed to the use of the use of more software agents for the search for free resources in the clusters of the HPC system. These mobile agents will be of the autonomous type and controlled by the same server. The mobility of the agents will have the same function as the Lévy flights.

## 11. Conclusions

This paper presents the way in which the parallelization research work of the array method has evolved, the main objective of which is to improve the performance of the two phases of a scheduler in an HPC system through parallel programming techniques.

The first research work presented the array method. The second work presented the first phase of parallelization, using OpenMP libraries to make use of shared memory programming and to speed up the intra-node execution of the algorithm; this work presents the parallelization of the array method with OpenMP and MPI hybrid programming for shared memory programming and distributed memory programming which improves performance in a server farm; a non-sequential resource search method is presented, using Lévy random walks, which allows searching for resources in different geographically distributed clusters. The meta-heuristic used to find the best assignments of the tasks in the clusters is parallelized.

In this work the metrics makespan, waiting time, quality of assignments and search for free resources were evaluated. The performance of the sequential algorithm and the parallel algorithm was compared. By parallelizing the phases of the array method such as the extraction of features from the PTGs, search for assignments in the clusters with the meta-heuristic used and the parallelization of the meta-heuristic used, better results

in the total execution times of the tasks are obtained, the waiting times of the tasks in the waiting queue are reduced, a greater number of free resources dispersed in the clusters is obtained with the search algorithm that uses Lévy random walks, so that the quality of the assignments is significantly improved, and the execution of the search algorithm for free resources is carried out more frequently and in a greater number of clusters of the target system, allowing scanning in random geographically distributed positions.

**Funding:** This research was funded by Tecnológico Nacional de México grant number 13434.21-P. https://www.tecnm.mx (accessed on 30 September 2021).

**Data Availability Statement:** Not applicable.

**Acknowledgments:** Special thanks to Instituto Tecnológico el Llano Aguascalientes for administrative and technical support.

**Conflicts of Interest:** The author declares no conflict of interest.

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
