# Peer review of "Parallelization of Array Method with Hybrid Programming: OpenMP and MPI"

_applsci, doi:10.3390/app12157706_

Round 1
Reviewer 1 Report
The author propose an investigation related to the parallelization of array method with hybrid programming: OpenMP and MPI. The topic of the paper is actual and interesting. The article is suitable for the Special Issue "Applications of Parallel Computing" of Applied Sciences Journal. The structure of the paper is well prepared. I have the following remarks:
- page 3,line 112 -"results are describen" have to be "results are described"
- page 4, line 159 - "...workloads W constituted.." ->'"...workloads W constituted.."
- page 5, line 202 - "...their controllers.in [31]..." have to be "...their controllers. In [31]..."
- I think that part of the section 4 have to be moved to the section 1.
- The text of the figures is merged with the text after them - it can be added a row between them.
Author Response
According to your revision I would like to comment:
all remarks was modified,
part of the section 4 was moved to the section 1
and finally, due to format of MDPI I can't separate the text of the figures, I am waiting for response of a mail that I sent to editors of the journal about how can I separate this paragraphs
Reviewer 2 Report
Author addresses an important problem, however the analysis presented in the manuscript seems to be quite simplified and not detailed enough. Only simulation results without detailed theoretical discussion OR implementation.
Full implementation of such a system may be a big task, however a small scale implementation can be used for testing OR a more comprehensive analysis can be done.
Author Response
According to your revision I would like to comment:
a more comprehensive analysis was made, into Introduction section and related works section,

Reviewer 3 Report
Major comments:
The reviewed paper proposes an interesting problem, which is not only theoretically, but could be also practically important. Authors focused on shared memory programming; distributed memory programming; Hybrid Programming; High Performance Computing Systems; Clusters; Array Method; Parallel Task Graphs.
Authors study an parallelization of applications with high processing times and large amounts of storage in High Performance Computing (HPC) systems, shared memory programming and distributed memory programming have been used; a parallel application is represented by Parallel Task Graphs (PTGs) using Directed Acyclic Graphs (DAGs). For the execution of PTGs in HPC systems, a scheduler is executed in two phases: secheduling and allocation; the execution of the scheduler is considered an NP-Complete combinatorial problem and requires large amounts of storage and long processing times. Array Method (AM), is a scheduler to execute the task schedule in a set of clusters; this method was programmed sequentially, analyzed and tested using real and synthetic application workloads in previous work. Analyzing the proposed designs of this method in this research work, the parallelization of the method is extended using hybrid OpenMP and MPI programming in a server farm, at the same time, a novel method for searching free resources in clusters using Levy random walks is proposed. Synthetic and real workloads have been experimented with to evaluate the performance of the new parallel schedule and compare it to the sequential schedule. The metrics of Makespan, waiting time, quality of assignments and search for free resources were evaluated; the results obtained and described in the experiments section show a better performance with the new version of the parallel algorithm compared to the sequential version.
The paper has a logical structure and is clearly, concisely and accurately written.
I would suggest to update Abstract and Conclusions to highlight most important findings of this research.
The Introduction part should be updated, the authors did not clearly show the difference between their approach and those in the literature. Complex and expanded state-of-the-art is needed.
I also suggest author should add discussion about pros and cons of considered problem to clearly identify the benefits of the introduced approach.
It would be intersting for readers if the paper include theoretical section about extended number of different potential practical applications of presented approach in different areas. I would suggest to add it.
I strongly suggest to add all [1]-[2] appropriate references from the list below:
[1] https://doi.org/10.1142/S0129626418500160
[2] https://doi.org/10.3390/app9173509
Minor comments:
Paper contains some amount of typos that need to be corrected throughout the paper. There are several minor language errors in the text. Some sentences require rewriting.
Author Response
According to your revision I would like to comment:
Abstract and Conclusions was rewritten to highlight most important findings of this research work were made,
Introduction section was updated,
pros and cons subsection was written,
Two references, proposed by you was added,
Paper was revised by the authors, and some sentences was rewriting.

Round 2
Reviewer 2 Report
I recommend the authors to share their source code over GitHub.
Reviewer 3 Report
The current version of the paper "Parallelization of Array Method with Hybrid Programming: OpenMP and MPI" contains the required materials as suggested and it is acceptable for publication in the Journal.
I have no further suggestions regarding the improvements of the content.
Abstract: For parallelization of applications with high processing times and large amounts of storage 1 in High Performance Computing (HPC) systems, shared memory programming and distributed 2 memory programming have been used; a parallel application is represented by Parallel Task Graphs 3 (PTGs) using Directed Acyclic Graphs (DAGs). For the execution of PTGs in HPC systems, a scheduler 4 is executed in two phases: secheduling and allocation; the execution of the scheduler is considered 5 an NP-Complete combinatorial problem and requires large amounts of storage and long processing 6 times. Array Method (AM), is a scheduler to execute the task schedule in a set of clusters; this 7 method was programmed sequentially, analyzed and tested using real and synthetic application 8workloads in previous work. Analyzing the proposed designs of this method in this research work, 9 the parallelization of the method is extended using hybrid OpenMP and MPI programming in a 10 server farm and using a set of clusters geographically distributed; at the same time, a novel method 11 for searching free resources in clusters using Levy random walks is proposed. Synthetic and real 12 workloads have been experimented with to evaluate the performance of the new parallel schedule and 13 compare it to the sequential schedule. The metrics of Makespan, waiting time, quality of assignments 14 and search for free resources were evaluated; the results obtained and described in the experiments 15 section show a better performance with the new version of the parallel algorithm compared to the 16 sequential version. By using the parallel approach with hybrid programming applied to the extraction 17 of characteristics of the PTGs, applied to the search for geographically distributed resources with 18 Levy random walks and applied to the metaheuristic used, the results of the metrics are improved. 19 The Makespan is decreased even when the loads increase, the times of the tasks in the waiting queue 20 are decreased, the quality of assignments in the clusters is improved by causing the tasks with their 21 subtasks to be assigned in the same clusters or in clusters neighbors and finally the searches for free 22 resources are executed in different geographically distributed clusters, not sequentially. 23
Keywords: shared memory programming; distributed memory programming; Hybrid Programming; 24 High Performance Computing Systems; Clusters; Array Method; Parallel Task Graphs
Conclusions 615
This paper presents the way in which the parallelization research work of the array 616 method has evolved, the main objective of which is to improve the performance of the two 617 phases of a scheduler in an HPC System through parallel programming techniques. 618
A first research work presented the Array Method. A second work presented the 619 first phase of parallelization, using OpenMP libraries to make use of shared memory pro- 620 gramming and to speed up the intranode execution of the algorithm; this work presenta la 621 parelizacion del Array Method with OpenMP and MPI hybrid programming for shared 622 memory programming and distributed memory programming which improves perfor- 623mance in a server farm; a non-sequential resource search method is presented, using Levy 624 random walks, which allows searching for resources in different geographically distributed 625 clusters. The metaheuristic used to find the best assignments of the tasks in the clusters is 626 parallelized. 627
In this work the metrics: Makespan, waiting time, quality of assignments and search 628 for free resource were evaluated. The performance of the sequential algorithm and the 629 parallel algorithm was compared. By parallelizing the phases of the array method such 630 as the extraction of features from the PTGs, search for assignments in the clusters with 631 the metaheuristic used and the parallelization of the metaheuristic used, better results in 632 the total execution times of the tasks are obtained, the waiting times of the tasks in the 633 waiting queue are reduced, a greater number of free resources dispersed in the clusters is 634
Number of resources found
Version April 12, 2022 submitted to Journal Not Specified 19 of 20
obtained with the search algorithm that uses Levy’s random walks, so that the quality of 635 the assignments is significantly improved, the execution of the search algorithm for free 636 resources is carried out more frequently and in a greater number of clusters of the target 637 system, allowing scanning in random geographically distributed positions.